

# Identification and characterization of *Streptomyces flavogriseus* NJ-4 as a novel producer of actinomycin D and holomycin

Zhaohui Wei, Chao Xu, Juan Wang, Fengxia Lu, Xiaomei Bie and Zhaoxin Lu

College of Food Science and Technology, Nanjing Agricultural University, Nanjing, Jiangsu, China

## ABSTRACT

This paper is the first public report that *Streptomyces flavogriseus* can produce both actinomycin D and holomycin. The actinomycete strain NJ-4 isolated from the soil of Nanjing Agricultural University was identified as *S. flavogriseus*. This *S. flavogriseus* strain was found for the first time to produce two antimicrobial compounds that were identified as actinomycin D and holomycin. GS medium, CS medium and GSS medium were used for the production experiments. All three media supported the production of actinomycin D, while holomycin was detected only in GS medium and was undetectable by HPLC in the CS and GSS media. The antimicrobial activity against *B. pumilus*, *S. aureus*, *Escherichia coli*, *F. moniliforme*, *F. graminearum* and *A. niger* was tested using the agar well diffusion method. Actinomycin D exhibited strong antagonistic activities against all the indicator strains. Holomycin exhibited strong antagonistic activities against *B. pumilus*, *S. aureus* and *E. coli* and had antifungal activity against *F. moniliforme* and *F. graminearum* but had no antifungal activity against *A. niger*. The cell viability was determined using an MTT assay. Holomycin exhibited cytotoxic activity against A549 lung cancer cells, BGC823 gastric cancer cells and HepG2 hepatocellular carcinoma cells. The yield of actinomycin D from *S. flavogriseus* NJ-4 was 960 mg/l. *S. flavogriseus* NJ-4 exhibits a distinct capability and has the industrial potential to produce considerable yields of actinomycin D under unoptimized conditions.

# INTRODUCTION

To date, several actinomycete genomes have been sequenced and annotated. Among actinomycetes, the genus *Streptomyces* remains a focus of systematic research and is especially of major pharmaceutical interest based on its commercial value as a rich source of numerous secondary metabolites.

*Streptomyces* species are soil bacteria, from which numerous antibiotic and antifungal compounds have been derived (*Weber, 2015*). *Streptomyces flavogriseus* has been reported to produce several enzymes, such as exoglucanase, extracellular proteases, cellulase, xylanase, and glucose isomerase (*Chen, Anderson & Han, 1979*; *Mackenzie, Bilous & Johnson, 1984*;

Corresponding author
Zhaoxin Lu, fmb@njau.edu.cn

*Ghorbel et al., 2014*), and antibiotics, such as the polyketide antitumour antibiotic xantholipin, the polycyclic xanthone bromoxantholipin, epithienamycin and xanthomycin (*Chen et al., 2011*; *Zhang et al., 2012*; *Arasu, Duraipandiyan & Ignacimuthu, 2013*). However, no public data have been found demonstrating that *S. flavogriseus* can produce actinomycin D or holomycin.

Actinomycins are a very famous class of chromopeptides produced by many species of *Streptomyces* and *Micromonospora* (*Kurosawa et al., 2006*; *Wagman et al., 1976*). All known actinomycins possess the same phenoxazone chromophore but differ in the amino acid composition of the two pentapeptidolactone side chains. Among the actinomycins, actinomycin D accounts for a significant number of drugs, including antibiotics and antitumour agents, and the clinical value of this actinomycin has prompted numerous studies over the last century. Since actinomycin D was first isolated from *Streptomyces antibioticus* (*Waksman, Geiger & Reynolds, 1946*), early studies mainly concentrated on the preparation, biochemical properties and pharmaceutical mechanism of this compound. Actinomycin D has been clinically used to treat Wilms' tumour (*Green, 1997*) and childhood rhabdomyosarcoma (*Womer, 1997*). This compound has also been recommended for the treatment of acquired immunodeficiency syndrome based on the ability of actinomycin D to inhibit human immunodeficiency virus minus-strand transfer and insertion into double helical DNA, thereby inhibiting the activity of the DNA-dependent RNA polymerase (*Rill & Hecker, 1996*). Although investigation has slowed recently, there are still some studies that focus on relevant issues of production, such as the yield, environment (*Rebecca & Peter, 2014*) and optimum fermentation conditions (*Praveen, Tripathi & Bihari, 2008*; *Praveen et al., 2008*). This continued study indicates that it not only has traditional clinical value but also has significance in continuing academic research.

Holomycin is an antibiotic compound with a dithiolopyrrolone structure. This compound was first discovered in *Streptomyces griseus* (*Ettlinger et al., 1959*) and later was isolated from *Streptomyces sp*. P6621 (*Okamura et al., 1977*) and mutants of *Streptomyces clavuligerus* (*Kenig & Reading, 1979*). The gram-negative bacteria *Photobacterium halotolerans* S2753 (*Wietz et al., 2010*) and *Yersinia ruckeri* (*Qin et al., 2013*) have also been reported to be holomycin producers. The gene cluster for holomycin biosynthesis was first reported in *S. clavuligerus* ATCC 27064 (*Li & Walsh, 2010*) and later in *Pseudoalteromonas sp*. SANK 73390 (*Fukuda et al., 2011*) and *Y. ruckeri* (*Qin et al., 2013*). Holomycin has antimicrobial activity against the filamentous fungi *Mucor miehei* and most bacteria, including rifamycin-resistant bacteria, but holomycin has no antimicrobial activity against *Saccharomyces cerevisiae* (*Oliva et al., 2001*; *Cui et al., 2006*). The antibacterial mechanism of holomycin is still unclear, but holomycin inhibits RNA synthesis *in vivo* in *Escherichia coli* (*Kenig & Reading, 1979*; *Webster, Li & Chen, 2000*; *Guo, Chen & Bin, 2008*).

In this report, we describe the characteristics of the *Streptomyces* strain NJ-4, which was identified as *S. flavogriseus* based on morphological, cultural, physiological, and biochemical characteristics as well as molecular methods. We found that *S. flavogriseus* NJ-4 could produce two well-known antibiotics, actinomycin D and holomycin, when grown on Gause's synthetic medium. *S. flavogriseus* NJ-4 also produced a large amount

of actinomycin D in CS medium under unoptimized conditions. This is the first report showing that *S. flavogriseus* can produce actinomycin D and holomycin.

## MATERIALS AND METHODS

### Strains and cell lines

Actinomycete strain NJ-4 was isolated from the soil of Nanjing Agricultural University Campus in Nanjing, China. The method of isolating strain NJ-4 was as fllows: a soil sample (5 g) was suspended in 45 ml of 0.85% physiological saline solution. The soil suspension was incubated at 28 °C with shaking for 30 min and then allowed to settle. 100 µl of the suspension was coated evenly on Gause's synthetic (GS) agar medium (*Huang et al., 2008*) consisting of soluble starch 2.0%, $KNO_3$ 0.1%, $K_2HPO_4 \cdot 3H_2O$ 0.05%, NaCl 0.05%, $MgSO_4 \cdot 7H_2O$ 0.05%, $FeSO_4 \cdot 7H_2O$ 0.001% and agar 2.0% (pH 7.0∼7.2) and incubated at 28 °C for 7∼10 days. The isolate was purified and stored on GS agar medium.

*Bacillus pumilus* CMCC 63202 was obtained from the National Center For Medical Culture Collection; *Staphylococcus aureus* ATCC 25923 and *Escherichia coli* ATCC 25922 were obtained from the American Type Culture Collection; *Fusarium moniliforme* CGMCC 3.4017, *Fusarium graminearum* CGMCC 3.4598 and *Aspergillus niger* CGMCC 3.6478 were obtained from the China General Microbiological Culture Collection Center. Bacteria was cultured on nutrient agar medium (*Ichimiya et al., 1996*) at 37 °C for 24 h. Fungi was cultured on PDA agar medium (*Huang et al., 2008*) at 30 °C for 7 days.

A549 lung cancer cells (*Maruyama & Majerus, 1985*), BGC-823 gastric cancer cells (*Wang et al., 2012*), HepG2 hepatoma cells (*Knowles, Howe & Aden, 1980*) and 293T normal human cells (*Bogerd et al., 2014*) were collected by Laboratory of Enzyme Engineering, Nanjing Agricultural University. All the cells were cultured in RPMI 1640 medium (*Wang et al., 2012*), supplemented with 10% (v/v) bovine calf serum at 37 °C with 5% $CO_2$, and 95% air.

### Characterization and identification of strain NJ-4

Strain NJ-4 was first identified by morphological, cultural, physiological, and biochemical characteristics according to the programmes recommended by Shirling and Gottlieb (*Shirling & Gottlieb, 1966*). Pridham and Gottlieb carbon utilization medium (PG medium) was used to test the utilization of carbon sources. After autoclaving the PG medium, the medium was cooled to approximately 50 °C, and a sterile carbon source was added to give a concentration of approximately 1%. The mixture was then agitated, and plates were immediately poured. Mature spores (50 µl) were coated evenly onto the surface of the solidified agar plates. PG medium without a carbon source was used as a negative control, and PG medium plus glucose was used as a positive control. The growth of strain NJ-4 was observed after 7–10 days of incubation at 30 °C. Mature substrate mycelium and aerial mycelium pigmentation were recorded on several types of ISP agar media after incubating at 30 °C for 14 days. The NaCl tolerance, pH range, and temperature range for growth were recorded on Gause's synthetic agar plates, which were incubated at 30 °C for up to 14 days.

Strain NJ-4 was then identified by homology analysis of its 16S rDNA gene. Genomic DNA was extracted using a Bacterial DNA Kit (OMEGA, USA) according to the methods described by the manufacturer. The 16S rDNA gene was amplified by PCR using the universal primers 27f (5′-AGAGTTTGATCTGCCTCAG-3′) and 1492r (5′-TACGGYTACCTTGTTACGACTT-3′). The PCR programme consisted of initial denaturation at 94 °C for 4 min, followed by 35 cycles at 94 °C for 30 s, 55 °C for 30 s, and 72 °C for 2 min, with a final extension for 10 min at 72 °C. The PCR product was purified using a PCR product purification kit (Shanghai Sangon Biotech, Shanghai, China), cloned into the pMD 19-T cloning vector (Dalian Takara, China), and then transformed into chemically competent *E. coli* DH5α (*Hanahan, 1983*). Sequencing was performed at the Invitrogen of Shanghai sequencing facility. The homology analysis of the 16S rDNA was performed by BLAST in GenBank (NCBI), and a neighbour-joining phylogenetic tree was constructed using MEGA 5.0 software.

## Purification and characterization of actinomycin D and holomycin

Strain NJ-4 was grown on a Gause's synthetic agar slant at 30 °C for seven days, and the mature spores were transferred into a 250-ml flask containing 50 ml of Gause's synthetic medium. The seed culture was incubated at 30 °C with shaking at 180 rpm for 2 days. For holomycin and actinomycin D production, 5 ml of seed culture was transferred into a 250-ml flask containing 50 ml of Gause's synthetic medium, which was incubated at 30 °C with shaking at 180 rpm for seven days.

The cultured broth was centrifuged at $8,000 \times$ g for 15 min after seven days of cultivation, and the supernatant and mycelia were extracted three times with ethyl acetate at room temperature. The ethyl acetate fractions were combined, concentrated, and dried under reduced pressure to give a red-orange powder. The red-orange powder was then dissolved in methanol and loaded onto a Sephadex LH-20 column ($2.6 \times 120$ cm). Elution was carried out with 80% methanol at a flow rate of 0.3 ml/min. Fractions were collected with a fraction collector, and antimicrobial activity was assessed in all fractions using the agar well diffusion method. Two fractions with antimicrobial activity were obtained. The two fractions were separately evaporated to dryness under vacuum and further purified by preparative RP-HPLC (Waters XBridge Prep C18 OBD column, 5 μm, $150 \times 19$ mm, Waters Delta 600). The first fraction was suspended in methanol, and 100 μl of the concentrated first fraction was then loaded onto the column and separated using an isocratic gradient of 70% acetonitrile in water as the mobile phase at a flow rate of 4 ml/min for 20 min. The purified first fraction was collected automatically according to its absorbance at 443 nm. The second fraction was purified in a manner similar to that used for the first fraction but with an isocratic gradient of 25% acetonitrile in water as the mobile phase for 12 min and UV detection at 386 nm. The purified fractions were characterized using ultraviolet light (SHIMADZU UV-2600 UV-Vis spectrophotometer), infrared light (Thermo Mattson Fourier transform infrared spectrophotometer), electrospray ionization mass spectrometry (Thermo Electron Corporation, San Jose, CA, USA) and nuclear magnetic resonance (NMR) spectroscopy (BRUKER).

## Production of actinomycin D and holomycin in different fermentation media by strain NJ-4

Three types of fermentation media were used: (1) GS medium consisting of the same medium as in Gause's synthetic agar medium except for agar (pH 7.0∼7.2); (2) CS medium consisting of corn flour 2.0%, soybean flour 1.5%, glucose 0.05%, yeast extract 0.025% and $CaCO_3$ 0.01%; (3) GSS medium consisting of soluble starch 2.0%, soybean flour 1.5%, $KNO_3$ 0.1%, $K_2HPO_4 \cdot 3H_2O$ 0.05%, NaCl 0.05%, $MgSO_4 \cdot 7H_2O$ 0.05% and $FeSO_4 \cdot 7H_2O$ 0.001%. Every 24 h for seven days, 2 ml of culture was collected and prepared for HPLC analysis as follows: After centrifugation ($8,000 \times$ g, 15 min), the supernatants and mycelia were extracted twice with ethyl acetate. The two ethyl acetate fractions were combined and dried under reduced pressure. The crude extract was then re-dissolved in methanol, filtered through a 0.22- μm nylon membrane filter, and analysed with an RP-HPLC (DIONEX Ultimate 3000) instrument equipped with an Agilent Eclipse XDB-C18, 5- μm column ($4.6 \times 250$ mm), with UV detection at 443 nm and 70% acetonitrile in water as the mobile phase for actinomycin D or with UV detection at 386 nm and 20% acetonitrile in water as the mobile phase for holomycin.

## Antimicrobial activity assay

Antimicrobial activity against *B. pumilus*, *S. aureus*, *E. coli*, *F. moniliforme*, *F. graminearum* and *A. niger* was tested using the agar well diffusion method (*Perez, Paul & Bazerque, 1990*). Nutrient agar medium and PDA agar medium were melted in a microwave, and after cooling to approximately 50 °C, the respective indicator microorganism was added at $10^7$ cfu/ml, and the plates were immediately poured. When the agar plates had solidified, wells of 5 mm in diameter were cut using a cork borer. The wells were filled with 20 μl of the purified and commercial actinomycin D and holomycin at a concentration of 1 mg/ml in methanol. The inhibition zones were detected using callipers after 24 h of incubation at 37 °C for bacteria and after 48 h of incubation at 30 °C for fungi.

## MTT assay for cell viability

A549 lung cancer cells, BGC-823 gastric cancer cells, HepG2 hepatoma cells and 293T normal human cells were dispensed into 96-well plates and treated with holomycin for 24 h. The cell viability was then determined using an MTT assay according to the procedure described by Price and McMillan (*Price & McMillan, 1990*). Absorbance was measured at 570 nm using a microplate spectrophotometer (Thermo LabSystems, Milford, MA, USA).

## Determination of oxidation reduction potential and NAD(+)H

Strain NJ-4 was grown in GS medium, CS medium and GSS medium. Every 24 h for 7 days, 5 ml of culture was collected and centrifuged at $8,000 \times$ g for 15 min at 4 °C. Oxidation reduction potential of the supernatant was determined by a pH/mV meter (SevenEasy Plus, METTLER TOLEDO, Switzerland). The mycelia were washed three times with ice-cold PBS buffer. NAD+ and NADH were assayed using the NAD(+)H determination kit (Comin Biotechnology Co., Ltd., Suzhou, China) according to the methods described by the manufacturer.

**Table 1  Culture characteristics of strain NJ-4.**

| Strain | Medium | Growth | Aerial mycelium color | Substrate mycelium color |
|--------|--------|--------|-----------------------|--------------------------|
| NJ-4 | ISP2 | +++ | Dark-gray | Green-yellow |
| | ISP3 | +++ | Leaden-gray | yellow |
| | ISP4 | +++ | gray | Gray-yellow |
| | ISP5 | ++ | gray | Orange-yellow |
| | ISP6 | ++ | gray | Light-yellow |
| | ISP7 | ++ | Light-gray | yellow |

Notes.

+++, heavy growth; ++, moderate growth.

## Statistical analysis

All assays were done in triplicate and results were expressed as the mean ± standard deviation (SD). Results were statistically analyzed by ANOVA (Analysis of variance) by SAS 9.0 software.

## RESULTS

### Identification of the isolated strain NJ-4

The cultural and physiological characteristics of strain NJ-4 are summarized in Tables 1 and 2, respectively. All the traits of strain NJ-4 were nearly the same as those of *S. flavogriseus* (*Chen, Anderson & Han, 1979*; *Ghorbel et al., 2014*) except for the production of actinomycin D and holomycin. The GenBank accession number of the 16S rDNA gene of strain NJ-4 is KM102731. By comparing the 16S rDNA gene sequence of strain NJ-4 with the sequence in GenBank, this strain was found to have a high similarity with *S. flavogriseus*. Strain NJ-4 formed a phylogenetic cluster with *S. flavogriseus* and other *Streptomyces* (*S. caviscabies*, *S. praecox*, *S. pratensis*, *S. flavofuscus*, *S. anulatus*, *S. fimicarius*) in the phylogenetic tree according to a phylogenetic analysis (Fig. 1). We believe that strain NJ-4 should be classified as the conventional species of *S. flavogriseus* on the basis of phylogenetic analysis and physiological properties.

### Purification and characterization of actinomycin D and holomycin from *S. flavogriseus* NJ-4

Two compounds with antimicrobial activity were separated from the antimicrobial crude extract by Sephadex LH-20. The first compound with antimicrobial activity was isolated as a red-orange powder and was further purified by HPLC with an HPLC retention time of 14.8 min (Fig. S1). The first component with antimicrobial activity had a typical UV-Vis spectrum in methanol with maximal absorbance peaks at 241 nm (shoulder) and 443 nm, similar to actinomycin D (Fig. S2). The IR spectrum (KBr) indicated the presence of $-C=O$ (1746.23 cm$^{-1}$ and 1646.91 cm$^{-1}$) and $-NH$ (3445.21 cm$^{-1}$ and 3273.57 cm$^{-1}$). There were bands at 2874.38 cm$^{-1}$ and 2965.02 cm$^{-1}$ because of the symmetrical and asymmetrical C–H stretching of the $-CH_2$ group (Fig. S3). ESI-MS of the first compound with antimicrobial activity revealed an intense ion at m/z 1255.22 [M+H]$^+$ and 1277.38 [M+Na]$^+$, which was identical to the results observed with actinomycin D
**Table 2  Physiological and biochemical characteristics of strain NJ-4 and related species.**

| Test items | NJ-4 | *S. flavogriseus* | *S. caviscabies* | *S. praecox* | *S. pratensis* | *S. flavofuscus* | *S. anulatus* | *S. fimicarius* |
|---|---|---|---|---|---|---|---|---|
| Growth on sole carbon source (1%, w/v) | | | | | | | | |
| D-Glucose | + | + | + | + | + | + | + | + |
| L-Arabinose | + | + | + | − | + | − | + | + |
| D-Xylose | + | + | + | + | + | + | + | + |
| D-Fructose | + | + | + | + | + | + | + | + |
| L-Rhamnose | + | + | + | + | − | + | + | − |
| D-Mannitol | + | + | + | + | + | − | + | + |
| D-Sucrose | − | − | − | − | − | − | − | − |
| D-Raffinose | − | − | − | − | − | − | − | − |
| meso-Inositol | − | − | − | + | − | − | − | + |
| Hydrolysis of starch | + | + | + | + | + | + | + | + |
| Liquefaction of gelatin | + | + | + | + | − | + | − | + |
| Peptonization of milk | + | + | + | + | + | + | + | + |
| Reduction of nitrate | − | − | + | + | + | + | + | + |
| H$_2$S production | − | − | − | − | − | − | − | − |
| Melanoid pigment | − | − | − | + | − | − | − | − |
| pH range of growth | 6~10 | 5~10 | 6~11 | 5~10 | 6~11 | 5~10 | 5~10 | 5~10 |
| Temperature range of growth | 10~37 °C | 10~37 °C | 10~37 °C | 10~37 °C | 10~37 °C | 10~37 °C | 10~37 °C | 10~37 °C |
| NaCl (4%) | + | + | + | + | + | + | + | + |
| NaCl (7%) | − | − | + | − | + | − | − | − |

**Notes.**
+, growth; −, no growth.

(*Kurosawa et al., 2006*) (Fig. 2). As shown in Table 3, the [1]H and [13]C NMR spectrum of the compound were consistent with the previously reported NMR spectrum of actinomycin D (*Chen et al., 2012*). The structure of actinomycin D was shown in Fig. S4. Based on all the above data, the first compound with antimicrobial activity was determined to be actinomycin D.

The second compound with antimicrobial activity was isolated as orange-yellow prisms and was further purified by HPLC with an HPLC retention time of 9.4 min (Fig. S5). The UV spectrum of the second compound with antimicrobial activity exhibited a maximum peak at 386 nm in methanol, which is characteristic of a pyrroline ring (Fig. S6). The IR spectrum of the second active component was identical to that published for holomycin (Fig. S7) (*Fuente et al., 2002*). ESI-MS of the second active compound revealed an intense ion at m/z 237.0 [M+Na]$^+$, which was the same as that observed with holomycin (Fig. 3). The $^1$H and $^{13}$C NMR data of the second active component (500 MHz, DMSO-d$_6$, $^1$H NMR: 2.02 ppm, s; 7.04 ppm, s; 9.83 ppm, s; 10.67 ppm, s; $^{13}$C NMR: 22.31 ppm, 110.45 ppm, 115.36 ppm, 133.69 ppm, 133.92 ppm, 167.87 ppm, 168.78 ppm) were the same as the NMR spectrum of holomycin (*Wietz et al., 2010*). The structure of holomycin was shown in Fig. S8. Based on all the above information, the second active compound was identified as holomycin.

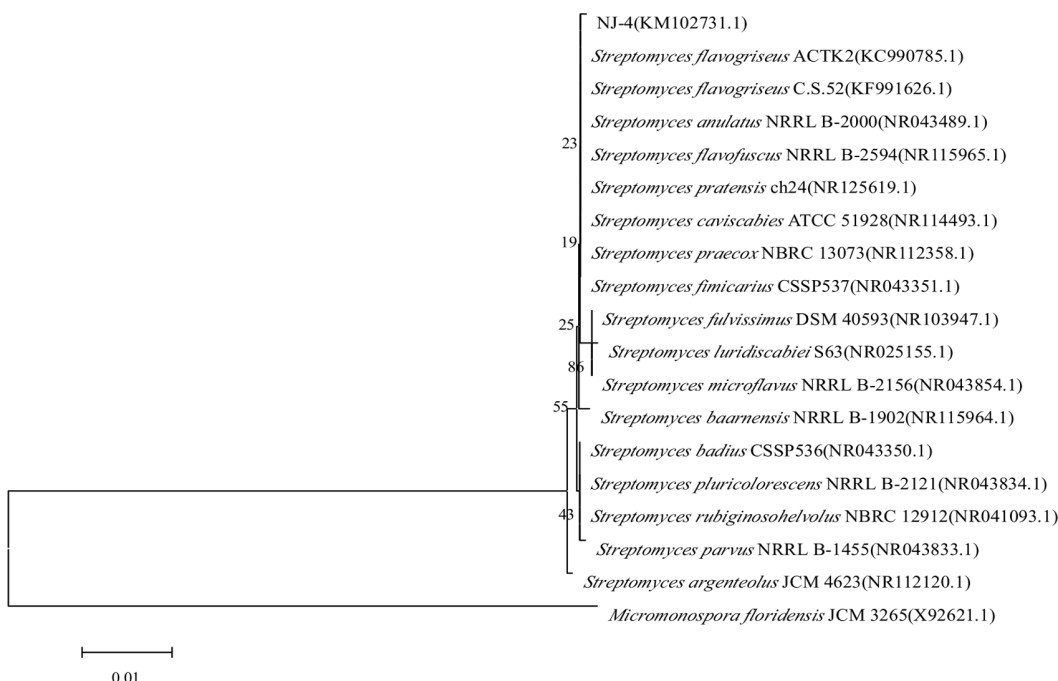

**Figure 1  Neighbor-joining phylogenetic tree of strain NJ-4 based on 16S rRNA gene sequence generated by MEGA 5.0.** Numbers at nodes indicate levels of bootstrap support based on a neighbor-joining analysis of 1,000 resampled datasets. The scale bar (0.01) indicates the number of nucleotide substitutions per site. NCBI accession numbers are given in parentheses.

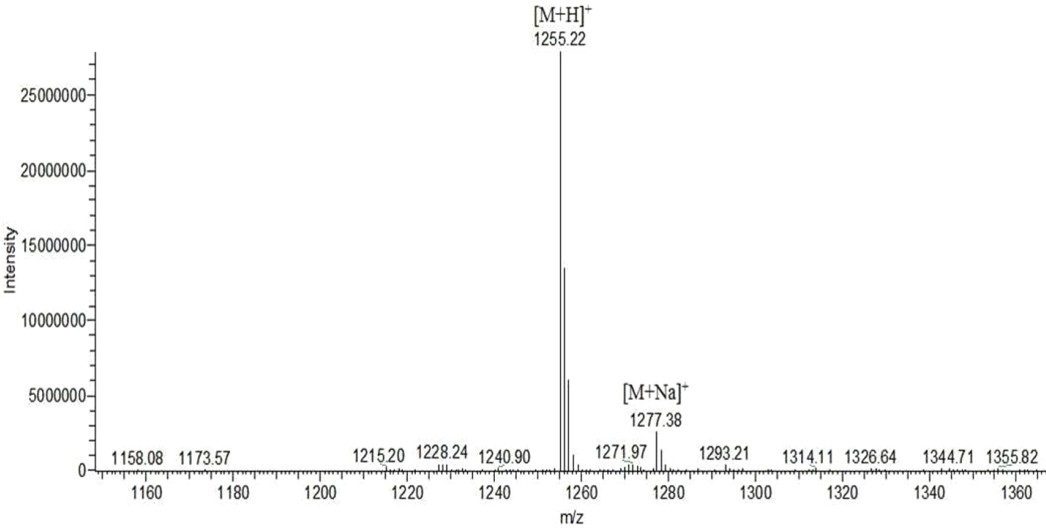

**Figure 2  Mass spectrometry analysis of the first active compound.** ESI-MS of the first antimicrobial activity compound revealed an intense ion at m/z 1255.22 [M+H]$^+$ and 1277.38 [M+Na]$^+$, which was the same as actinomycin D.
**Table 3  NMR data of actinomycin D (CDCl3, $^1$H: 500 MHz, $^{13}$C: 75.4 MHz).**

| α-ring | Position | $\delta_C$ | $\delta_H$ | β-ring | Position | $\delta_C$ | $\delta_H$ |
|---|---|---|---|---|---|---|---|
| Thr | 1 | 168.51 | | Thr | 1 | 167.55 | |
| | 2 | 55.26 | 4.62 (dd) | | 2 | 54.90 | 4.52 (dd) |
| | 3 | 75.05 | 5.21 (qd) | | 3 | 74.98 | 5.18 (qd) |
| | 4 | 17.76 | 1.26 (3H, d) | | 4 | 17.34 | 1.25 (3H, d) |
| | NH | | 7.16 (d) | | NH | | 7.76 (d) |
| D-Val | 1 | 173.74 | | D-Val | 1 | 173.32 | |
| | 2 | 58.90 | 3.56 (dd) | | 2 | 5873 | 3.53 (dd) |
| | 3 | 31.82 | 2.11 (m) | | 3 | 31.55 | 2.07 (m) |
| | 4 | 19.26 | 1.13 (3H, d) | | 4 | 19.08 | 1.12 (3H, d) |
| | 5 | 19.03 | 0.91 (3H, d) | | 5 | 18.98 | 0.89 (3H, d) |
| | NH | | 8.15 (d) | | | | 8.01 (d) |
| Pro | 1 | 173.37 | | Pro | 1 | 173.32 | |
| | 2 | 56.44 | 6.03 (d) | | 2 | 56.26 | 5.96 (d) |
| | 3 | 31.29 | 1.89, 2.65 (m) | | 3 | 30.97 | 1.85, 2.65 (m) |
| | 4 | 23.00 | 2.15, 2.23 (m) | | 4 | 22.85 | 2.15, 2.23 (m) |
| | 5 | 47.59 | 3.74 (2H, dd) | | 5 | 47.33 | 3.84 (2H, m) |
| Sar | 1 | 166.53 | | Sar | 1 | 166.32 | |
| | 2 | 51.39 | 4.82, 3.64 (d) | | 2 | 51.39 | 4.71, 3.60 (d) |
| | NMe | 34.91 | 2.91 (3H, s) | | NMe | 34.85 | 2.88 (3H, s) |
| MeVal | 1 | 167.65 | | MeVal | 1 | 166.53 | |
| | 2 | 71.45 | 2.68 (m) | | 2 | 71.28 | 2.68 (m) |
| | 3 | 26.92 | 2.68 (m) | | 3 | 26.92 | 2.68 (m) |
| | 4 | 21.66 | 0.96 (3H, d) | | 4 | 2,156 | 0.95 (3H, d) |
| | 5 | 19.21 | 0.75 (3H, d) | | 5 | 19.08 | 0.74 (3H, d) |
| | NMe | 39.27 | 2.94 (3H, s) | | NMe | 39.14 | 2.88 (3H, s) |

Chromophore

$\delta_H$ 2.56 (3H, s), 2.68 (3H, s), 7.37 (d), 7.64 (d)

$\delta_C$ 7.74, 15.00, 101.75, 113.52, 125.80, 127.59, 129.13, 130.25, 132.65, 140.50, 145.12, 145.90, 147.64, 166.53, 168.99, 179.12

## Production of actinomycin D and holomycin by *S. flavogriseus* NJ-4 in different fermentation media

Actinomycin D was detected in the GS medium, CS medium and GSS medium and was eluted at 14.8 min during HPLC. The production of actinomycin D began within 24 h in three fermentation media, with a maximum yield of actinomycin D of 960 mg/l produced in CS medium over 7 days (Fig. 4). To our knowledge, approximately 30 species of *Streptomyces* and *Micromonospora* have been reported to be capable of producing actinomycin D (*Kenig & Reading, 1979*; *Kurosawa et al., 2006*).

Holomycin was detected only in the GS medium, with a maximum holomycin yield of 9.16 mg/l in GS medium (Fig. 5). Holomycin was not detected when *S. flavogriseus* NJ-4 was grown in CS medium or GSS medium.

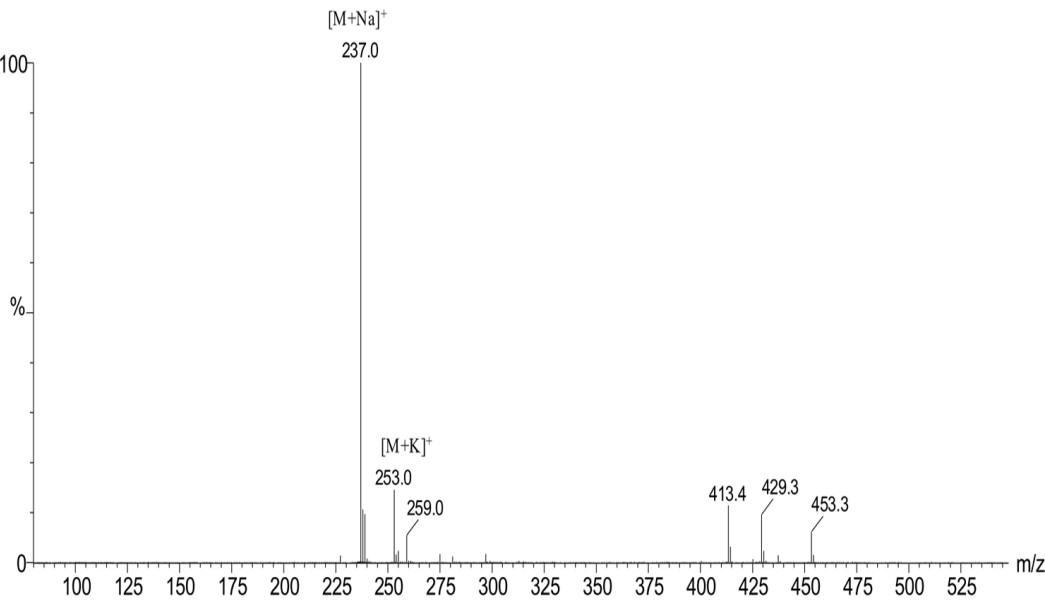

**Figure 3  Mass spectrometry analysis of the second active compound.** ESI-MS of the second active compound revealed an intense ion at m/z 237.0 $[M+Na]^+$ and 253.0 $[M+K]^+$, which was the same as holomycin.

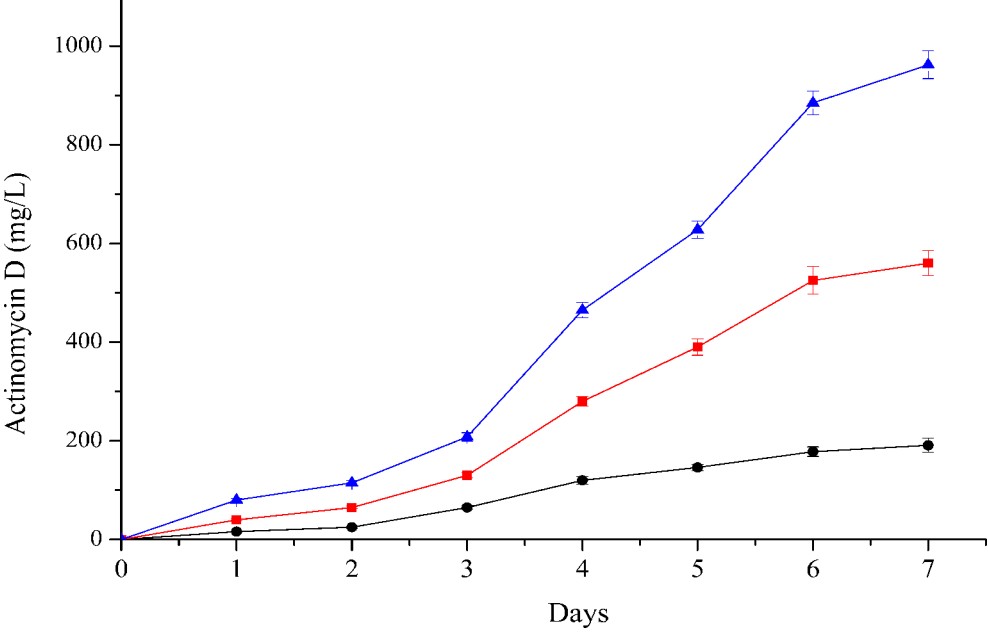

**Figure 4  Time course production of actinomycin D by strain NJ-4 cultured in CS (triangle), GSS (square) and GS (circle) media.** Actinomycin D was detected in CS, GSS and GS media. Production of actinomycin D started within 24 h in three fermentation media, with the maximum yield of actinomycin D production 960 mg/L in CS medium over seven days.

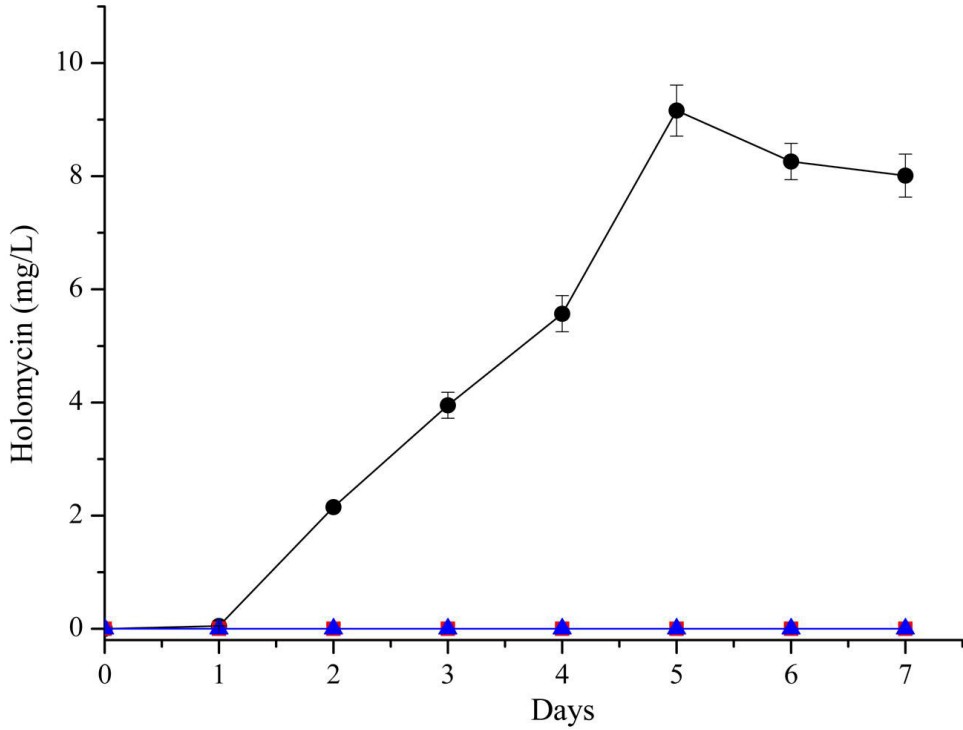

**Figure 5** **Time course production of holomycin by strain NJ-4 cultured in CS (triangle), GSS (square) and GS (circle) media.** Holomycin was only detected in GS medium, with the maximum yield of holomycin 9.16 mg/l in GS medium. Holomycin was not detected by HPLC in CS medium and GSS medium.

**Table 4** **Antimicrobial activity of actinomycin D and holomycin.**

| Test organism | Diameter of inhibition zone (mm)[a] | | | | | |
|---|---|---|---|---|---|---|
| | *E. coli* | *B. pumilus* | *S.aureus* | *F. moniliforme* | *F.graminearum* | *A.niger* |
| Actinomycin D[b] | $34.73 \pm 0.22$ | $32.42 \pm 0.28$ | $33.56 \pm 0.35$ | $27.46 \pm 0.14$ | $28.69 \pm 0.37$ | $28.06 \pm 0.19$ |
| Actinomycin D[c] | $34.58 \pm 0.29$ | $32.89 \pm 0.34$ | $33.06 \pm 0.19$ | $27.35 \pm 0.34$ | $28.62 \pm 0.12$ | $27.55 \pm 0.41$ |
| Holomycin[b] | $29.13 \pm 0.21$ | $27.83 \pm 0.18$ | $29.94 \pm 0.15$ | $15.52 \pm 0.39$ | $16.52 \pm 0.12$ | – |
| Holomycin[c] | $28.92 \pm 0.34$ | $27.68 \pm 0.39$ | $29.73 \pm 0.32$ | $15.56 \pm 0.14$ | $16.35 \pm 0.28$ | – |

**Notes.**

[a] The value represents the mean $\pm$ standard deviation of triple determinations.

[b] Actinomycin D and holomycin purified from strain NJ-4.

[c] Commercial actinomycin D and holomycin purchased from Sigma-Aldrich and Toronto Research Chemicals respectively.

–, no antifugal activity.

## Antimicrobial activity assay

As shown in Table 4, purified and commercial actinomycin D exhibited the same strong antagonistic activities against all the indicator strains (*E. coli*, *B. pumilus*, *S. aureus*, *F. moniliforme*, *F. graminearum* and *A. niger*). Purified and commercial holomycin exhibited the same strong antagonistic activities against *E. coli*, *B. pumilus* and *S. aureus* and had antifungal activity against *F. moniliforme* and *F. graminearum* but had no antifungal activity against *A. niger*.

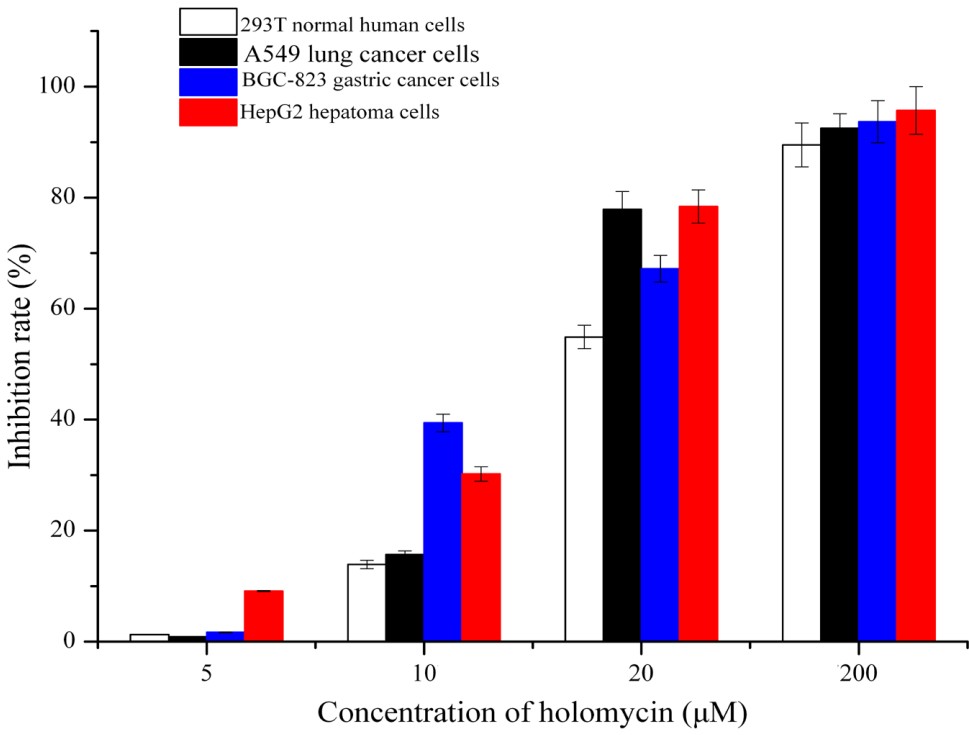

**Figure 6  Proliferative inhibition of holomycin on 293T normal human cells, A549 lung cancer cells, BGC-823 gastric cancer cells, HepG2 hepatoma cells.** Research results indicate that holomycin not only has antitumor activity against A549 lung cancer cells, BGC-823 gastric cancer cells, and HepG2 hepatoma cells, but also has cytotoxic activity against 293T normal human cells. More than half of all the cancer cells was inhibited with the holomycin concentration at 20 μM.

## Effects of holomycin on the viability of eukaryotic cells

The data obtained from the MTT assay showed that holomycin exhibited cytotoxic activity against all three human tumour cell lines. When the concentration of holomycin was 5 μM, the inhibition of all cancer cells and 293T normal human cells was weak. When the concentration of holomycin was 10 μM, the inhibitory effect on BGC-823 gastric cancer cells and HepG2 hepatoma cells became obvious, while the viability of all cancer cells was inhibited by over 50% at 20 μM holomycin (Fig. 6).

Research results indicate that holomycin not only has antitumour activity against A549 lung cancer cells, BGC-823 gastric cancer cells, and HepG2 hepatoma cells but also has cytotoxic activity against 293T normal human cells.

## DISCUSSION

### Identification of the strain *S. flavogriseus* NJ-4 and its novel products of actinomycin D and holomycin

The search for novel antibiotics for pharmaceutical, industrial and agricultural applications has never ceased and is of immense importance worldwide, especially the search in unexplored habitats for antibiotics that are effective against resistant pathogenic microorganisms. Since the full genome sequence analysis revealed novel features compared

to other *Streptomyces* strains, the *Streptomyces* genus has been widely used as an important biological tool for producing novel antibiotics (*Medema et al., 2010*). Therefore, the isolation, identification and evaluation of the strain *S. flavogriseus* NJ-4 obtained from a soil sample from the campus of Nanjing Agricultural University, China is of great significance.

We found that *S. flavogriseus* NJ-4 can produce both actinomycin D and holomycin when grown in Gause's synthetic medium. *S. flavogriseus* NJ-4 also produced a large amount of actinomycin D in CS medium under unoptimized conditions.

Approaches allowing the characterization of the remaining gene clusters will undoubtedly hold potential for uncovering the biosynthetic pathways for other known or unidentified metabolites from *S. flavogriseus*. However, there are still no reports that *S. flavogriseus* can produce actinomycin D and holomycin. Although many specimens of *Streptomyces* isolated from soil have been reported to produce actinomycin D or holomycin (*Kenig & Reading, 1979*; *Kurosawa et al., 2006*; *Huang et al., 2011*), *S. flavogriseus* NJ-4 is the first reported strain producing both actinomycin D and holomycin.

## Production of actinomycin D and holomycin by *S. flavogriseus* NJ-4 in different fermentation media

Only a few strains have been reported to produce relatively large quantities of actinomycin D, including *S. griseoruber* (210 mg/l actinomycin D) (*Praveen & Tripathi, 2009*), *S. parvulus* (152 mg/l actinomycin D under optimized conditions) (*Praveen, Tripathi & Bihari, 2008*), a mutant strain of *S. sindenensis* (850 mg/l actinomycin D) (*Praveen et al., 2008*) and *S. avermitilis* (1,770 mg/l actinomycin) (*Chen et al., 2012*). Our *S. flavogriseus* NJ-4 strain exhibited the distinct capacity to produce a considerable amount of actinomycin D with a production of 960 mg/l over seven days. The production of actinomycin D by *S. flavogriseus* NJ-4 is greater than that of other actinomycin D producers, except *Streptomyces avermitilis*. These results suggest that *S. flavogriseus* NJ-4 may be a better producer of actinomycin D. We have reason to believe that *S. flavogriseus* NJ-4 has the potential for industrial application after optimization of the culture conditions.

The production of holomycin by *S. flavogriseus* NJ-4 was undetectable in CS medium and GSS medium. The biosynthesis of holomycin was probably strongly affected by the redox level of the cells (*Nardiz et al., 2011*). In CS medium and GSS medium, soybean flour as an organic nitrogen source reduces the redox level of the fermentation broth and the *S. flavogriseus* NJ-4 cells (Figs. S9 and S10), which might inhibit holomycin biosynthesis.

If we use *S. flavogriseus* NJ-4 as an industrial producer of actinomycin D and holomycin, we must consider optimizing the conditions of the industrial production of these compounds. To steadily increase the yield and enhance the purity, further studies are required regarding the nurturing environments and impacts of the process .

## Antimicrobial activity and viability of cancer cells

Actinomycin D exhibited strong antimicrobial activity in our study. As a well-known clinical antitumour drug, the effects of actinomycin D on the viability of cancer cells, including A549 lung cancer cells, BGC-823 gastric cancer cells and HepG2 hepatoma cells, have been widely investigated (*Newman, Flower & Croxtall, 1994*; *Singhal & Rajeswari,*

*2009*; *Li et al., 2013*). Therefore, for this article, the effect of actinomycin D on the viability of eukaryotic cells was not tested.

Holomycin exhibited antifungal activity against two species of plant pathogenic fungi, including *F. moniliforme* and *F. graminearum*. These results do not conflict with the results of other studies. Generally, holomycin, a member of the pyrrothine class of antibiotics, is a valuable lead compound for the development of agricultural fungicides. Cui has reported on the antifungal activity of holomycin against *Mucor miehei* (*Cui et al., 2006*).

As a well-known product of various strains of *Streptomyces*, holomycin is a member of the structural class of dithiolopyrrolones. Dithiolopyrrolone antibiotics generally have broad-spectrum antibacterial activity against various microorganisms, including Gram-positive and Gram-negative bacteria, and even parasites (*Tanner et al., 1950*). It has been proposed that holomycin is a prodrug that requires intracellular conversion into an active species, which then inhibits RNA polymerase (*Oliva et al., 2001*). Holomycin inhibits RNA synthesis *in vivo* in *E. coli* (*Kenig & Reading, 1979*; *Webster, Li & Chen, 2000*; *Guo, Chen & Bin, 2008*), but the antibacterial mechanism of holomycin against bacteria is still unclear.

As an antibiotic with a dithiolopyrrolone structure, holomycin possesses antimicrobial activity against bacteria and fungi (*Oliva et al., 2001*; *Cui et al., 2006*). Dithiolopyrrolone compounds, including holomycin, have been reported to be drugs that are cytotoxic to several mammalian cell lines (*Jia et al., 2010*). In our research, holomycin exhibited inhibitory activity against 293T normal human cells as well as the three tested cancer cell types. The structural modification of holomycin to minimize cytotoxic activity against normal human cells and to investigate the structure-function relationship in this class of compounds is ongoing (*Chen et al., 2003*).

## CONCLUSION

*S. flavogriseus* NJ-4 was found to produce two well-known antibiotics, actinomycin D and holomycin. However, holomycin could be detected only in the GS medium and was not detected in the CS medium or the GSS medium. The biosynthesis of holomycin is probably strongly affected by the redox level of the cells (*Nardiz et al., 2011*). The mechanism by which holomycin production is affected by cellular redox levels will be further elaborated in our next study. Further research on holomycin should be conducted to minimize its cytotoxic activity against normal human cells, to investigate the relationship between the structure and function of this class of compounds and to increase the holomycin yield.

As a better producer of actinomycin D, *S. flavogriseus* NJ-4 exhibits a distinct capability and the industrial potential to produce considerable amounts of actinomycin D under unoptimized conditions. To meet the needs of industrial pharmaceuticals, further study is required on the improvement of the industrial process environment, as well as the purity of the products, in the pursuit of mass production, especially for actinomycin D.

## ACKNOWLEDGEMENTS

We thank Zhe Song (China Pharmaceutical University) for measuring the nuclear magneti resonance (NMR) spectra and infrared spectra of actinomycin D and holomycin. We thank Zhihong Xin (Nanjing Agricultural University) for spectral analysis.

### Funding

This work was supported by the National Science and Technology Support Program (grant 2011BAD23B05) of the People's Republic of China. The funders had no role in study design, data collection and analysis, decision to publish, or preparation of the manuscript.

### Grant Disclosures

The following grant information was disclosed by the authors:
National Science and Technology Support Program: 2011BAD23B05.

### Competing Interests

The authors declare there are no competing interests.

### Author Contributions

- Zhaohui Wei conceived and designed the experiments, performed the experiments, analyzed the data, wrote the paper, prepared figures and/or tables, assist other matters not mentioned above.
- Chao Xu performed the experiments.
- Juan Wang performed the experiments, assist other matters not mentioned above.
- Fengxia Lu and Xiaomei Bie analyzed the data, contributed reagents/materials/analysis tools, reviewed drafts of the paper.
- Zhaoxin Lu conceived and designed the experiments, reviewed drafts of the paper.

### DNA Deposition

The following information was supplied regarding the deposition of DNA sequences:
The GenBank accession number of the 16S rDNA gene of strain NJ-4 is KM102731.

### Data Availability

The raw data has been uploaded as a Supplementary File.

### Supplemental Information

Supplemental information for this article can be found online at http://dx.doi.org/10.7717/peerj.3601#supplemental-information.

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
