# Peer review of "Identification and characterization of Streptomyces flavogriseus NJ-4 as a novel producer of actinomycin D and holomycin"

_PeerJ, doi:10.7717/peerj.3601_

## Round 0.1 · original submission · Major Revisions

As you can see, all 3 reviewers had serious concerns about the manuscript. As we aim to maintain a high standard of scientifically sound findings, I strongly recommend you to respond to every one of the reviewer´s comments and modify as required.

Reviewer 1 ·

Basic reporting

After reviewing the manuscript I strongly recommend to the authors get editing help from someone with full professional proficiency in English. Considering that the language used is not sufficiently comprehensible and needs to be improved, since due to shortcomings in the language of the manuscript, I could not fully assess its quality and was difficult to read.

The authors should include actual references in order to improve this section.

Methods section needs major corrections. Particullarly, it should be important authors describe an experimental design, e.g. number of experiments, replicates, statistical analysis.

A major concerns along the text is that authors fail to discuss why their aim was: isolate and identify a Streptomyces strain NJ-4 with high antibiotic production capability by morphological, cultural, physiological, biochemical characteristics and molecular methods. Their discussion was ambiguous.

The literature cited into the text should not be appropriately cited. Multiple references to the same item should be separated with a semicolon (;) and ordered chronologically.

Experimental design

The present manuscript comply with the aims and scope of the journal. However, the research drafted in the present form is ambiguous. Authors declare that "The aim of the present study was to isolate and identify a Streptomyces strain NJ-4 with high antibiotic production capability by morphological, cultural, physiological, biochemical characteristics and molecular methods". Some important questions are proposed: 1) Site for sampling? 2) Methodology used for actinomycetes isolation? 3) criteria to select a specific strain? 4) use of control metabolites actinomycin D and holomycin to compare antimicrobial activity and analytical study, 5) number of assays? for antimicrobial activity against bacteria and fungi and how any replicates?, 6) In general method section lacks of clearity.

Validity of the findings

METHODS
Molecular methods: Indicate the DNA polymerase enzyme used for gene amplification.
Describe the molecular criteria used to implement phylogenetic tree construction. Selection of sequence, editing of sequence to analyzing the 16S DNA gene, why the authors do not use an outgroup sequence to tree construction?
Metabolites purification: The following phrase is confused, lines 114-117: All isolatable fractions were collected by fraction collector for the antagonistic activity assay. Two antimicrobial activity fractions were obtained. The antimicrobial activity fractions were pooled, concentrated, and gave a pure active constituent. The authors must clarify the complete paragraph.

DISCUSSION
Lines 210-211 and 215-217: The idea is duplicated in the same paragraph¡¡¡¡ (Although approaches allowing the characterization of the rest of the gene clusters will undoubtedly hold potential for uncovering biosynthetic pathways for other known or unidentified metabolites from Streptomyces flavogriseus).

Ambiguous paragraph: Lines 238-241, “On the antimicrobial activity and on the viability of cancer cells Actinomycin D has exhibited strong antagonistic activities in our research. In line with its consistent performance as a compound of actinomycetes which have provided a large amount of the naturally occurring antibiotics extensively used by the pharmaceutical industry”
Lines 252-254, “Holomycin inhibit RNA synthesis in vivo in Escherichia coli (Guo et al. 2008; Kenig and Reading 1979; Webster et al. 2000), but the antibacterial mechanism of holomycin against
bacteria is still unclear”. This same idea is repeated at lines 256-257
Taxonomic mistake: lines 242-243, “And for holomycin, it has exhibited antifugal activity against two species of plant pathogenic “bacteria” as F. moniliforme and F. graminearum”
Paragraph with different ideas: Lines 246-249, “Streptomyces microorganisms possess many biosynthetic gene clusters for secondary metabolites, but many of them are silent. It is not clear what the function of many gene products is, nor the identity of the biosynthetic metabolites of these silencing genes. As a well-known product of various strains of streptomyces, holomycin is a structural class of dithiolopyrrolones”. In this case the authors must reorganize their thoughts.

Additional comments

Multiple improvements must be done to the manuscript, some of them are follows:
ABSTRACT
Line 26-28. Confused paragraph
INTRODUCTION
Check phrasing and spelling
Line 23: And holomycin exhibited strong antagonistic activities against B. pumilus, S….. Check the previous sentence in connection with this
Line 42: Actinomycins is a very famous class. What mean´s this expression?
Line 46: Since it was first isolated from Streptomyces antibioticu(s). Check the correct spelling name
Line 34-35: Phrase: “but also for the taxonomic difficulties within the genus caused by the large number of isolates and insufficient species definition” is confuse. Taxonomic description of strains is an important task for researchers especially for the discovery of novel species producers of new bioactive metabolites, for this reason a polyphasic criteria is actually used to describe novel species, e.g physiological, biochemical, molecular, morphological traits. Please redraft this idea¡¡
REFERENCES
The authors do not follows the format suggested in the instructions to cite the references.

Reviewer 2 ·

Basic reporting

The manuscript by Zhaohui Wei et al reports a new strain of Streptomyces flavogriseus. The fact that this strain is able to produce both actinomycin D and holomycin makes it unique among the hitherto known strains of S. flavogriseus. Moreover, the amount of actinomycin D produced by this strain makes it a good candidate for a future production of the drug.
I think the data shown in the paper is consistent and supports the conclusions. The article structure is correct. However the language needs strong revision and improvement. I found several places with awkward wording, grammar mistakes or sentences difficult to understand. A few examples:
- line 161. “…were because of symmetrical and asymmetrical…”
- line 45. “actinomycin D has accounts…”
- line 236. Holomycin, it needs to do further study on…” should read “further study is needed…”
- line 255. “holomycin has possessed antagonistic activity” should be read “holomycin possesses antagonistic activity”
- line 267-268. “The toxic safety is the basic clinical research requirements that we must understand its inherent mechanism especially for holomycin”
- line 242. After a full stop the sentence starts like this “And for holomycin…”, which makes it sound a bit colloquial.
-etc…

Moreover, in the Introduction section, a space is needed before and/or after many references.

Some references are needed in the Material and Methods section:
l.77. Gause’s Synthetic Agar
l.84. Nutrient agar medium
l. 84. PDA agar medium
l. 86. Reference for cell lines used
l. 87. Reference or recipe for RPMI 1640 medium
l. 101. DH5 alpha strain
l. 103. NCBI should be mentioned
Also in the Results and Discussion sections:
l. 163. “which was the same as actinomycin D” (ref?)
l. 249-250 “…Gram-positive and –negative bacteria, and even parasites”


Some panels of the same figure do not share the same pattern (Fig. S1 and S2). The font and the size are different and some axes are in bold. Also, some figures contain an (a) or (b) inside the graphic and also outside (Figures 2, 3, S1, S2 and S3). Please, modify appropriately.

Figure 1 legend should mention what the numbers at the nodes mean and authors should also state that the numbers given in parenthesis are the NCBI accession number.

Experimental design

Methods are described with sufficient detail. However, I have some specific comments:
- A few references are missing (see above).

- In “Characterization and identification of strain NJ-4”, the authors do not explain how was the utilization of carbon sources tested (from Table 2). Please, define the procedure and the concentrations used.

- In “Production of actinomycin D and holomycin in different fermentation media by strain NJ-4”: the authors say they collect cultures every 24h, they should also mention for how long they do so (7 days). In line 128, if first day is collected, it should read: “Every 24h, 2 ml of culture was collected”.

- In “Antimicrobial activity assay”: the antimicrobial activity of the fractions collected was tested by the agar diffusion method and this is well explained. However, if I did not misunderstand table 4, the assay is carried out with purified actinomycin D and holomycin. This should be explained in this section and the concentrations of both drugs should be mentioned.

Validity of the findings

A concern for the reviewer is in the first section of results.
First, if one performs a Blastn alignment of sequence KM102731 with all the 16S rRNA of the NCBI database, other species of Streptomyces come up with a bit more homology than S. flavogriseus. Why did the authors choose those species to perform the alignment? I have noticed that another actinomycin producing strain also appears in the alignment file (S. sindenensis). Have the authors tried to compare the physiological characteristics with some other actimomicyn D producing strains (for example S. sindenensis) or with some other Streptomyces strains with a more similar 16S rRNA? If not, this should be done to confirm that strain NJ-4 belongs to this species.

Line 163. …”which was the same as actinomycin D (Fig. 2A)”. The authors should provide a reference for that.

Figure 4 would look much better if shown as a dose-response curve, from which you can also calculate the IC50.

Why the effect of actinomycin D on the viability of eukaryotic cells has not been tested in this manuscript? If it has not been performed because it has already been performed by other authors, it should be mentioned in the discussion section.

Table 4 shows the Antimicrobial activities of both drugs against six bacterial and fungal strains. It is well known that those drugs are broad spectrum antibiotics. If the intention of the authors is to demonstrate that the produced and purified drugs retain antimicrobial activity, a positive control using commercial actinomycin D and holomycin should be included. Also, concentrations used in this assay are needed.

Paragraph starting on line 220 discusses the production of actinomycin D by S. flavogriseus NJ-4 compared to other strains. In almost all the other studies, the amount of actinomycin D is lower than the amount produced by S. flavogriseus NJ-4 strain. There is one exception found in strain S. avermitilis. The authors claim that the differences may be due to the incubation time, suggesting that S. flavogriseus NJ-4 may be a better actinomycin D producer. The authors do not discuss though that the different media used in both studies (Chen et al (2012) use MPG media) may account for such differences in the yield of actinomycin D. To confirm this statement, the authors should grow S. avermitilis with CS media for 7 days otherwise the affirmation should be modified or removed.

As stated in Peer J. author’s guide, conclusions should identify unresolved questions / gaps / future directions. The authors of the manuscript should focus on that instead of listing the results obtained once again.

Additional comments

The study is interesting and provides useful information. However, I recommend that someone carefully go through the entire text and improve the writing as much as possible as there are many incomprehensible sentences and grammar mistakes (which I will not specifically list one by one).

List of minor changes:

Abstract:
l. 26-27: ...”over 7 days in the flask by now...”: What does that mean “by now” here?
l. 27-28: A word is missing between “considerable” and “of actinomycin D”

Introduction:
l. 45: biochWemical should read biochemical
l. 65: “against bacteria” is redundant if you say antibacterial before
l. 68-69: Please, rephrase “The two novel productions should...”
l.67-70: There are many verb tenses in the same paragraph (past tense, modal verb and future verb), please try to standardize it.
l. 72. “Antibioticsm” should read antibiotics

Material and Methods:
l. 108: “Five ml of seed culture was transferred...”
l. 100: After a full stop, the sentence starts with a “After cultivated 7 days, the cultured broth...”. Please, rephrase.
l. 111: No need to say respectively if both, the supernatant and mycelia are extracted with the same solvent
l. 114: Could “All isolatable fractions” be changed by a “all fractions”?
l. 115: Sometimes the authors use antimicrobial activity and sometimes they use antagonistic activity. Is there any specific reason for that? The assay in the section is named “antimicrobial activity assay”, so maybe it would be better to use antimicrobial.
l. 115-119: For a better understanding, please, split the paragraph in two.
126: …” 0.025% and CaCO3…”
l. 130: Remove “respectively”
l. 137-139: Please, rephrase

Results:
l. 174: “NMR date” should read “NMR pattern/spectrum/…”
l. 179: …production of 960 mg/L…”
l. 181: This sentence is repeated (lines 178-179).
l. 183-184: “Holomycin was not detected when S. flavogriseus NJ-4 was grown on CS…”
l. 186: Please, mention here the indicator strains used
l. 189: “…on viability of cancer cells”: The authors also test the effect on non-tumoral strains, why did they use this title?

Discussion:
l. 204-206: Rephrase this sentence, please.
l. 208: “…also produced a large number of actinomycin D” should read “…a large amount/ quantity/…”
l. 215: Rephrase “If it allowed the characterization of the rest of the gene clusters, …”
l. 208: What do the authors mean when they say “actinomycin D and holomycin are different from the previous reports”?
l. 231: “For the undetectable of holomycin…”, please rephrase.
l. 232: Explain how the redox levels of the cells are affected to holomycin production. At which level the production of the antibiotic is reduced?
l. 236: “…further studies are needed….”
l. 256-257: This sentence is repeated (l. 252).
l. 259: “…the antitumor activity against 293T normal human cells…”. If 293T are non-tumoral cells, I would not say the antitumor activity.
l. 267-268: Please, rephrase

Conclusions:
l. 279: Please, replace “two newly” for “well-known anti-microbial compounds”. Maybe the authors should rephrase the sentence if what they want to state is that this is the first report that shows production of those antibiotics in a S. flavogriseus strain.
l. 282-283: This sentence says that the maximum yields of both drugs have been obtained in GS medium. Please correct the name of the medium as actinomycin D maximum yield was obtained with CS medium.
l. 291: “…hepatocellular carcinoma cells and 293T cells”

Author contributions section is missing.

Reviewer 3 ·

Basic reporting

The English language could be improved. There are many instances where the subject-verb agreement or verb tense needs to be corrected. Examples of unclear or incorrect English usage includes:
1. Line 32, 'while among them'
2. Line 33, 'systematics'
3. Line 42 'Actinomycins is'
4. Line 43 'phenoxasone' is incorrectly spelled
5. line 46 'antibioticu
6. line 47 'early researches' and 'biochWemical'
7. line 49, 'It was also been recommended'
8. line 51, 'inhibite'
9. line 53 "relevant segment fields of its deep structure analysis' does not make sense.

In contrast to the written portion, the figures are clear and professional. The only suggesion for improvement would be to add a figure with the structures of actinomycin and holomycin

The discussion is all over the place and many parts could be removed. For example, the entire paragraph talking about silent clusters is not relevant to the paper (obviously the actinomycin and holomycin gene clusters are not silent, since they are produced in large amounts)

Experimental design

The yields of the natural products needs to be more thoroughly described in the experimental. Was the production value determined with a single sample? How was the amount of natural product determined? Were crude extracts used to determine the amount?

Validity of the findings

Data could be better supported by using a positive controls. For example, actinomycin D is produced by several commercial (or freely available) strains, and they could be used for comparison. Furthermore, actinomycin D is commercially available and could serve as a positive control. For antibacterial activity, often a known antibiotic is used to ensure that the assay is being corrently performed.

Additional comments

The major limitation to understanding and interpreting the results is the use of the English language. A native English speaking colleague might improve this area. Also, the general topics that are discussed in the discussion need to be more focused to the results of paper

---

## Round 0.2 · accepted · Accept

I just encourage you to take a look at the comments from reviewer 2 and further edit your manuscript to address those comments (which can be done while in production).

Reviewer 1 ·

Basic reporting

A substantial improvements to the new manuscript was done

Experimental design

A substantial improvements to method sections was implemented by authors

Validity of the findings

With the substantial modifications the manuscript is now ready to be accepted

Additional comments

With the changes suggested and implemented the manuscript was improved

Reviewer 2 ·

Basic reporting

The manuscript by Zhaohui Wei et al reports a new strain of Streptomyces flavogriseus. The fact that this strain is able to produce both actinomycin D and holomycin makes it unique among the hitherto known strains of S. flavogriseus. Moreover, the amount of actinomycin D produced by this strain makes it a good candidate for a future production of the drug.
The data shown in the paper is consistent and supports the conclusions. The article structure is correct.
With the modifications added, I think the article is suitable for publication.
I only have minor comments:
- Abstract:
* I would move sentence in line 10 “The antimicrobial….diffusion method” before “Holomycin exhibited…” in line 14
*I would move “The cell viability….MTT assay” from line 12 to line 16 before “Holomycin exhibited…carcinoma cells”

- Materials and Methods:
*line 66: “… was suspend” should read “was suspended”
* I have a few concerns about some bibliography introduced. For example: HepG2 cells reference should be Knowles et al., 1980 instead of an article published last year. This is just one example, but I am afraid there are a few more of the newly introduced references. The same for DH5alpha, etc…

-Results:
* Line 189: “…in three crude extracts…”. Please, mention the name of the 3 media.
* Antimicrobial activity assay: The authors should state that the the purified actinomycin D performs as the commercial one in terms of antimicrobial activity. This would strengthen their results.
* Material and methods of the new supplementary figures should be added.

Experimental design

Correct

Validity of the findings

The authors have addressed all the questions the reviewer had properly.

Additional comments

The study is interesting and provides useful information. I think the manuscript has improved considerably.